# Best Practices for Notification of Unexpected, Violent, and Traumatic Death: A Scoping Review

**DOI:** 10.3390/ijerph20136222

**Published:** 2023-06-25

**Authors:** Diego De Leo, Josephine Zammarrelli, Giulia Marinato, Marta Capelli, Andrea Viecelli Giannotti

**Affiliations:** 1Australian Institute for Suicide Research and Prevention, Griffith University, Brisbane, QLD 4122, Australia; 2Slovene Centre for Suicide Research, Primorska University, 6000 Koper, Slovenia; 3De Leo Fund, 35137 Padua, Italy

**Keywords:** death notification, best practices, general guidelines, training programs, professional figures, telephone notification, notifying children

## Abstract

Background: Death reporting is a delicate task. The ways in which it is carried out can have a significant impact on both the recipient and the notifier, especially in the event of a sudden, violent, and traumatic death. Empathetic, sensitive, and attentive communication with survivors can represent a first opportunity to support the bereavement process. The acquisition of specific skills for the delivery of the death notification is necessary for the professional who carries out the communication to increase self-efficacy, knowledge, and perception of competence in this area. Objective: To map what the literature has produced on the theme of best practices for the notification of unexpected, violent, and traumatic deaths and to provide guidance for the formulation of appropriate best practices and the development of effective educational programs. Methods: A review was conducted using the PRISMA Scoping Review extension on English language literature published between 1966 and 2022. Results: Starting from the initial 3781 titles, 67 articles were selected. From a thematic point of view, the analysis of the contents made it possible to identify five dimensions: (1) general guidelines in relation to various professional figures; (2) specific protocols; (3) guidelines for notifying death to children; (4) guidelines for notification of death by telephone; and (5) recommendations and suggestions for death notification training programs. Discussion: Death notification is configured as a process, divided into sequential phases. The act of notification constitutes the central phase during which communication is carried out. The communication of death is context-specific; therefore, it should require the creation of specific protocols for the various professions involved in the task, along with targeted theoretical and practical training. Conclusions: The importance of defining specific guidelines for the various professionals and standardized programs of theoretical and practical training emerges. The implementation of future sectoral studies will allow evaluations of the effectiveness of these protocols and programs.

## 1. Introduction

Death communication is a delicate task that can have a significant impact on the emotional and behavioral responses of the recipient and on the entire bereavement process. In this article, the death of a loved one due to violent, unexpected, and external causes, such as suicide, homicide, accidents, or natural disasters, is defined as “traumatic death”. Experiencing this kind of loss can profoundly impact the bereavement process. Indeed, survivors are more likely to experience prolonged suffering, characterized by physical and psychological symptoms [1]. The methods by which notification of death is made assume particular importance in the event of a traumatic death. Numerous studies show that empathic, sensitive, and attentive communication with recipients can represent a first opportunity to support the victim’s survivors and prevent the development of pathologies such as persistent and complicated bereavement disorder or post-traumatic stress disorder [2,3,4,5].

The acquisition of specific skills and abilities for the delivery of notification of such a type of death would improve the experience of both the recipient and the professional who carries out the communication. In fact, the literature shows that the introduction of structured training programs in the professional growth path of health professionals, law enforcement agencies, and social workers increases self-efficacy, knowledge, and the perception of competence [2,6,7,8,9,10], consequently reducing the level of stress associated with death communication and the probability of burnout [11,12,13]. Years of work experience do not seem to be a sufficient condition to master the death notification task, so the introduction of guidelines for the different professions concerned would probably be of paramount importance to support the parties involved.

The aim of this scoping review is to provide an overview of the existing literature on best practices for notification of sudden, violent, and traumatic deaths. Although several scientific studies have proposed notification protocols, the need emerges to develop specific guidelines for the communication of traumatic deaths, taking into consideration the specific modalities of the death, the context, and the actors involved. The analysis here performed is not limited to emergency or clinical services but considers all contexts in which notification of traumatic death may be necessary. The secondary objective of this study is to provide indications for the formulation of best practices suitable for the situations previously described through the analysis of those studies that have targeted this highly challenging task.

## 2. Methods

The compilation of this review followed the PRISMA Extension Criteria for Scoping Review (PRISMA-ScR). All selected studies were reviewed, but no selection criteria were placed with respect to the outcome or design of the individual studies. There was no review protocol, and no registration was made.

Articles published in English from 1966 to the end of 2022 concerning the topic of best practices for death notification were selected. All articles appearing without abstracts or in the form of editorials or articles in periodicals, book reviews or book chapters, dissertations, and commentaries were excluded from the search. The review of the literature concentrated on the notification of death in cases of unexpected, violent, and traumatic deaths. In this sense, it refers only to external causes of death (i.e., accidents, suicides, and homicides).

This research focused on best practices for traumatic death notification. Therefore, we included only studies that explored: (a) general characteristics of ‘effective’ traumatic death notification (conceptual articles reporting guidelines, protocols, and best practices); (b) aspects specific to the death report or seeking to improve the death notification process (e.g., articles evaluating the validation and effectiveness of protocols; the setting; communication skills of the notifier, nonverbal language, gestures); and (c) death notification training programs (death notification training needs for specific professionals—e.g., police officers, emergency department personnel; role-playing, and simulations).

The definition of the eligibility criteria aimed at differentiating the material collected and dividing it into specific areas of investigation. In this research, articles reporting on non-traumatic and non-violent deaths were excluded. Deaths associated with palliative care were also excluded from the survey. The exclusion also concerned: (a) articles investigating perinatal death and infant death syndrome (SIDS); and (b) articles where the notification of death focused specifically on cases of cancer, cardiovascular, and neurological diseases.

Articles were searched in the following databases: EBSCO PsycInfo, EBSCO CINAHL, Scopus, Web of Science, and MEDLINE PubMed, using as keywords: “death communication”, “death notification”, communication and “traumatic death”, notification and “traumatic death”, communication and “sudden death”, notification and “sudden death”.

The authors of this work (J.Z., A.V.G., G.M., and M.C.) autonomously carried out bibliographic research for each keyword, as well as the subsequent elimination of duplicates. The same authors searched all databases. When checking for duplicates between different search engines, citations with a slightly different title and the same abstract, citations with the same title and abstract but a different year of publication, and citations with the same title and abstract but a different journal title were removed.

Once the final number of citations was obtained, three independent members of the research team (J.Z., A.V.G., and G.M.) performed the inclusion assessment in a standardized, open manner.

Disagreements among researchers were resolved with consensus methods. During the screening phase, it was not deemed necessary to review the full text of all articles. A fourth investigator (M.C.) completed the review process, while the senior author (D.D.L.) oversaw the entire review process. Figure 1 shows the document identification process according to the PRISMA-ScR flowchart.

The identification of the dimensions followed a phase of reflective and deductive thematic analysis. Once data collection was complete, four researchers on the team (J.Z., A.V.G., M.C., and G.M.) individually performed the analysis steps, during which they took notes on their initial impressions of each article. In a second step, the contents of interest (i.e., those in line with the research question) were assigned labels (a few words or a short sentence), which had the purpose of clearly evoking the relevant characteristics of the papers, in order to be able to encode them. Then, in full agreement with each other, the researchers defined a list of themes representing five dimensions. Those dimensions guided the subsequent research phases.

The risk of assessment bias was assessed at the study level. The primary risk of bias in the inclusion criteria was related to the decision, implemented prior to conducting the content analysis, to include studies where the type of death was not specified and studies evaluating death in relation to traumatic deaths but also other kinds of deaths. Furthermore, studies with different sizes and methodologies were also considered, resulting in a strong heterogeneity of the results. The risk of bias associated with the choice of articles in English was whether this language was associated with studies published faster and more often cited (see Figure 1).

## 3. Results

A total of 67 studies were selected for inclusion in our review. The analysis of study contents showed a considerable degree of similarity, with most studies of narrative type and conceptual content (i.e., the majority of the studies aimed to provide a global picture of the aspects involved in the notification process). Despite the inevitable overlapping among different articles, we identified five dimensions as representative of the main findings from the studies. The term “dimensions” indicates the main themes that emerged from the content analysis phase, i.e., a synthesis of information relating to a particular topic or data domain with shared meanings.

The dimensions identified are: (1) general guidelines in relation to various professional figures (*n* = 28); (2) specific protocols (*n* = 21); (3) guidelines for notifying children of a death (*n* = 11); (4) guidelines for telephone death notification (*n* = 21); and (5) recommendations and suggestions for death notification training programs (*n* = 32) (see Table 1). As anticipated, the total number of studies in each area exceeds the total number of selected studies due to the overlapping of several subject areas within the same study.

The notification of death in relation to the first dimension investigates the general guidelines, also in relation to various professional figures. It was necessary to make this specification because, from the textual analysis, three subgroups emerged: (1) general guidelines for various professional figures (*n* = 13); (2) guidelines for emergency department personnel (*n* = 10); and (3) law enforcement officers’ guidelines (*n* = 5).

The second area focuses on specific protocols in different contexts and experimental conditions (*n* = 21). Specific protocols for death notification are available in the Appendix A.

The guidelines for notifying children of a death constitute the third dimension. When analyzing the selected articles, we found suggestions on best practices for death communication to children and on strategies to support parents in this difficult task (*n* = 11).

Guidelines for notifying death by telephone do represent an area of study that is extremely timely, given the change in death notification procedures due to the COVID-19 pandemic (*n* = 21).

The fifth dimension analyzes the evidence that has emerged regarding death notification training programs: the need for training (attitudes, stressful content, work consequences, notification styles), educational courses/workshops, blended learning courses (education and experience), simulations, role-plays, and exercises (*n* = 32).

Overall, the evaluation of the 67 studies included in our review revealed the presence of a large heterogeneity between methodologies and research designs. The types of studies appeared to be distributed as follows: conceptual/narrative studies (*n* = 26); investigations (*n* = 6); cross-sectional investigations (*n* = 2); reviews (*n* = 4); qualitative studies (*n* = 9); pre-post design studies (*n* = 12); validation studies (*n* = 6); prospective observation (*n* = 1); and mixed method (qualitative/quantitative) design study (*n* = 1).

The prevalence of conceptual/narrative studies within our review and the paucity of quantitative studies did not allow us to aggregate data. It was not possible to trace the risk of bias in each individual study and assess the overall results or the results of each individual work.

### 3.1. First Dimension: General Guidelines and in Relation to Various Professional Figures

In the first dimension, a total of 28 studies were identified, divided into three sub-categories: thirteen articles provided general guidelines applicable to different professional figures; 10 studies were aimed at emergency department personnel; and five articles elaborated on specific guidelines for law enforcement agencies. The sub-categories identified have several elements in common: all articles reported phases or thematic areas to be studied in succession during the notification of death (contact with survivors, arrival of survivors, notification of death, response to pain, etc.) (*n* = 28); articles on selection of the staff and/or choice of a professional figure to assist the notifier (*n* = 9); articles about providing survivors with post-notification care and monitoring their condition through follow-up actions (*n* = 19); and, finally, articles protecting the privacy of survivors (*n* = 14). In this and subsequent dimensions, the total number of studies exceeds the total number of selected studies due to the overlapping of several subject areas among different studies. 

In the first sub-category, seven studies investigated some aspects specific to death notification personnel: understanding the emotional responses of survivors [21]; developing strategic plans to manage notification [39]; analyzing survivors’ privacy mechanisms regarding bereavement [52]; describing how professional roles and related emotional vulnerabilities can influence the process [35,66,67]; and the specific functions that professional roles can assume in different contexts, such as in the notification of a child’s death to the parents [29]. The other articles in this sub-category have focused on the type of death and the context of notification. Two studies have provided recommendations for the notification of death following road accidents, highlighting the need to draw up specific guidelines for this type of death [43] and to deepen the theme of the correct identification of the deceased and the information to provide about the accident [25]. One study presented a practical system for death notification and body identification for family members following a homicide, emphasizing the role of notifiers as first responders and providers of support for those experiencing this type of traumatic loss [38]. Research by Veilleux and Bilsky [54] provided several suggestions for notification procedures for mental health trainees following the suicide of a patient. Finally, Rivolta et al. [49] addressed the issue of communicating a patient’s death to residents of hospices and nursing homes (NH).

With respect to guidelines for emergency department personnel, several studies have investigated the role and actions of physicians during the death reporting process [14,17,18,23,32,44,46] including suggestions and practical procedures to be performed for body vision [1,14,17,19,23,46]. Further relevance was given to the choice of staff to carry out the notification and accompany the survivors throughout the process [1,14,19,44,46].

In the last subcategory, one study presented several factors that law enforcement officers and supervisors should consider in the notification process and offered guidelines to best serve agencies and communities [31]. The study by Hoffman et al. [64] provided further insight by analyzing the experiences of bereaved people who received death notifications from police officers. Research has highlighted the role of forensic nurses (FNDIs) as special reference figures to support investigations while taking care of survivors [50]. Reed et al. [59] examined interactions between homicide detectives and homicide co-victims as they unfolded through three predictable phases of the postmortem process: at the crime scene, during death notification, and throughout the investigation. Finally, the study by McGill et al. [3] explored the “knock on the door” notification process when notice is provided to the family by a member of the military (see Figure 2).

### 3.2. Second Dimension: Specific Protocols

The first death notification protocol found in our review was drafted by Moroni Leash in 1996 [20]. It is a model divided into eight sequential phases; according to the author, the notification process focuses on the following steps: time of contact, place of notification, arrival of family, questions for telephone notification, selection of the notifier, construction of the loss context, sequential notification technique, and vision of the body. Kaul [26] mentions the Moroni Leash protocol [20] and the 4-step Iserson model [69], designed in an abbreviated form to be easily incorporated into emergency room procedures. In his study, the author reviews the two protocols and proposes a version focused on the division of roles within the process [26].

To date, the most used protocol is the GRIEV_ING, acronym of a 9-step mnemonic procedure (Gather, Resources, Identify, Educate, Verify, Give Space, Inquire, Nuts and Bolts, and Give) to be performed during death notification within the emergency department [6,10,12,33,40,41,47,51,57,61,62,65].

In the United States, one of the most popular protocols is the so-called “Six-Step Protocol” for delivering bad news (SPIKES: Setting, Perception, Invitation, Knowledge, Empathy, Strategy, and Summary). It is primarily intended for clinicians providing information on poor prognoses, but it can also be used to inform family members of a patient’s death [7,8,58,60].

In addition to the use of SPIKES, Shomossi et al. [7] explored the familiarity of 97 Iranian nurses in communicating bad news, presenting further protocols: (1) the ABCDE, which proposes operational strategies through which to organize the notification action, such as “Active listening”, “Breathing retraining”, “Classification of needs”, “Direct to support networks”, and “psychoEducation”, and (2) the Stewart’s protocol [70], which suggests a three-step educational model centered on delivery modalities and the management of survivors’ and notifiers’ reactions. A study focuses on the “In Person, In Time” communication protocol, which provides useful information to effectively reduce the notifier’s stress level at the time of communication [8]. In addition, Sobczak has recently created the CONNECT protocol (acronym for Context, Organization, Near and Niceties, Emotions, Counseling, Taking care) to help healthcare professionals communicate bad news remotely through the use of the telephone [68] (see Figure 3).

### 3.3. Third Dimension: Guidelines for Death Notification to Children

The studies included in the third dimension have collected some fundamental aspects about death notification for children. The study by Dubin and Sarnoff [14] supports the need to encourage adult survivors to discuss deaths with children in order to reduce distressing fantasies, thus offering a model for expressing pain and facilitating the acceptance of the loss. For the same reason, it is necessary to support adults, through the assistance of childcare professionals, clergy, family members, or friends [39], within the notification process, in allowing children to see the body [20], making use of the help of the pediatric ward if in a hospital context [19]. It has been shown that children excluded from decision-making (e.g., body viewing, funerals) can report long-term psychological effects such as displays of anger, regret, and grief [3]. Therefore, it is important that children receive understandable and clear information [19] through honest and careful communication, which at the same time can comfort and protect them [52]. One study highlighted the potential of bibliotherapy as a tool to facilitate death communication for children who need to cope with the loss of a loved one [55].

In cases of homicide, it is recommended that the notification team evaluate the presence of children, supervise their reactions, and, possibly, decide to carry out the notification in a secluded place only with adults [38]. The EDECT seminar, for training emergency department physicians in death notification, devotes a specific section to ways of communicating with children [36]. Finally, the study by Servaty-Seib et al. [30] provided practical recommendations for schools and school counselors regarding preparation and follow-up associated with death notification situations (see Figure 4).

### 3.4. Fourth Dimension: Guidelines for Notification of Death by Telephone

In general, the results of our review suggest against notification of death by telephone unless relatives have to spend several hours reaching their loved one [14,16,17,20,23,25,26,27,31,35,38,54,59,66].

In emergency departments, most professionals prefer to use initial telephone contact with relatives of the deceased only to summon them to the hospital [25]. Although relatives believe they should be notified of the death when they are first called, several studies suggest that they later understand the reasons for deferring notification until their arrival [20,25,27,35].

In the event of homicide, if there is the possibility that the media will disseminate the identity of the victim, it is necessary for the investigators to notify the relatives as quickly as possible by telephone [59]. The study conducted by Stewart [25], also analyzed in other studies [35,38], provides practical recommendations for notification of death by telephone following a car accident or homicide. 

Following the COVID-19 pandemic, a study by Sobczak [68] formalized the creation of a specific protocol, called CONNECT, for communicating bad news remotely. The study by De Leo et al. [5] examined a further protocol, developed by Dyer [71], in which general guidelines for death notification are expanded, adding rules relating to telephone communication and appropriate timing. The remaining five studies highlighted the need for specific guidelines for telephone notification in hospital settings by proposing some standardized procedures [1,8,16,56,63] (see Figure 5).

### 3.5. Fifth Dimension: Best Practices for Death Notification Training Programs

We divided the studies related to this dimension into two sub-categories. One group investigated the current level of death notification training among various professionals, highlighting an educational gap and a high demand for death notification training by directly affected personnel [13,18,19,22,27,34,42,45,47,66]. In particular, it was noted the need to implement mnemonic protocols within specific teaching modules, explore the experiences of the actors involved, delivery styles, and emotion management strategies, raise awareness, and identify needs in order to prevent the negative impact of stressful contents from turning over and causing burnout. Only three studies targeted law enforcement agencies [15,59,64], two studies targeted various professionals [27,66], and the remainder were conducted within the hospital environment [13,18,19,22,34,42,45,47]. Finally, an article proposed the CADS measurement scale for assessing levels of communication anxiety and avoidance in the notifier as a useful tool for identifying training needs [53].

The second sub-category described the main training methods existing for death notification for various professional figures. In general, the results of this section have highlighted two different approaches: the use of theoretical modules supported by exercises and group discussions [5,11,24,27,36,46] or simulations and role-playing [2,9,10,16,28,41,48,58,60,61,63,65].

Specifically, a study analyzed the seminar developed by Mothers Against Drunk Driving (MADD) focused on learning the main actions related to the notification (e.g., correct selection of notifiers, identification of the deceased, contact with survivors, management of emotional reactions, etc.) and adaptable to different professional figures [27].

Two articles investigated the use of EDECT, an experiential training seminar aimed at changing emergency physicians’ attitudes and behaviors about dying that incorporates the curriculum for Continuing Medical Education (CME) related to the notification process of death in four steps [24,36]. Furthermore, Smith et al. [24] associated this seminar with Critical Incident Debriefing (CID), designed to mitigate the effects of traumatic stress on emergency services personnel. Brand and Mahlke’s [11] study discussed Death Notification with Responsibility (DNR), a blended learning course for police students on handling death notifications based on professional and legal requirements. Finally, one study proposed a series of exercises to promote a self-reflective approach to identifying the strengths and professional skills of emergency department trainees when caring for survivors [46].

The remaining studies have combined the theoretical modules with simulation as an opportunity to practice a notification method in controlled circumstances and through direct observation [28]. Simulation provides immediate and constructive feedback, which has been shown to be one of the most effective components of clinical [16] and interprofessional [48] teaching. Specifically, three studies used an educational program consisting of an introductory theoretical workshop and a simulation process based on the SPIKES protocol [41,58,60]. Instead, four articles have deepened the learning of the GRIEV_ING protocol through a theoretical training module and a simulation process accompanied by Rapid Cycle Deliberate Practice (RCDP), a technique that uses a series of immediate and repetitive debriefings to immediately identify if the feedback was understood and subsequently manifested in the simulation laboratory [10,61,62,65].

Finally, following the COVID-19 pandemic, some researchers have developed training programs that can be delivered through distance learning through training modules and simulations via video and telephone, developed on e-learning platforms [2,9,61,63] (see Figure 6).

## 4. Discussion

From the results of the review, a few observations emerged on the theme of best practices for the notification of unexpected, violent, and traumatic deaths.

One of the most obvious considerations is that notification of death is configured as a process divided into sequential phases, in which the act of notification simultaneously constitutes one of these steps. Consequently, the notification of death cannot be summed up in the communicative action itself but implies a more extensive and articulated procedure. Awareness of this evidence has important repercussions on practical recommendations for efficiently carrying out death notification and on training methods for selected staff. In other words, there is a need to be prepared for the act of notification and the notification process.

The act of notification is focused on the practical and material elements of the communication, such as, for example, the place, the clothing of the notifier, physical contact, the methods of explaining the event, the choice of staff, the timing, the availability, and the emotional control of the notifier. The notification process, on the other hand, begins before contact is made with relatives, continues during communication, and ideally ends with post-notification follow-up actions. The distinction between the act and the notification process is made clear by the analysis of the protocols for notification of death. Although they are presented in different forms, all protocols are focused on the notification act; they are presented in the form of acronyms to be easily memorized while summarizing the main actions to be carried out in the specific communication phase. However, all protocols involve their framing within a broader process: they presuppose an initial situation in which relatives have already been contacted and conclude with indications that ‘sow the seeds’ for actions to be taken following notification (e.g., body vision, organ donation, follow-up support actions, etc.).

A second relevant thematic area is the discovery that notification of death is necessarily context-specific. In other words, it is not possible to draw up training programs and generic guidelines for different professional figures. In fact, the present review demonstrates that generic guidelines are often incomplete; protocols are modified and readapted according to professional roles and different contexts, and, often, these facilitation maneuvers reduce both complexity and process accuracy. Not surprisingly, the most widespread and validated death notification protocol is the GRIEV_ING, which has been applied exclusively in medical-hospital settings. We have reason to suppose that context-specific analysis is hindered not only by practical difficulties in carrying out research in this area (e.g., survivor outcomes, longitudinal studies on lifelong learning, turnover, etc.) but also by numerous psychological aspects intrinsic to the notification process. Probably, in hospital settings, due to the work of multidisciplinary teams and the continuous contact with patients, the professional figures have greater confidence in the exploration of emotional psychological states such as avoidance, anxiety, and stress compared to other roles, such as providing military education to members of law enforcement agencies. To date, only five studies have explored guidelines for these professionals, despite this category being exposed, like doctors or nurses, to the notification process [3,31,50,59,64]. 

In emergency departments, staff typically contact relatives of the deceased by telephone, asking them to come to the hospital with a family member or close friend so that the communication can be made in person [14,17,46]. Emergency departments are often crowded and chaotic environments, so it would be convenient to have an isolated and silent room available [1,37], possibly near the recovery room [19].

Although every emergency department has its own internal organization regarding the choice of staff who will carry out the notification of death, the presence of the doctor who ascertained the death would be essential in this context [17,23]. In emergency departments, doctors are perceived as figures of reference by relatives waiting to receive news about their loved ones [14]. In these terms, he has the task of collecting all the information on the deceased person, organizing the chronology of the events that have occurred in clear and simple language, and answering all the questions of the relatives [14,17,23,44].

Against this background, relatives are very likely to ask to see the deceased. The doctor should encourage viewing of the body and provide some important information: where it is, any need for transport or further investigations (e.g., autopsy), and prepare survivors for changes to the body of the deceased (paleness, lesions, and redness). The ward staff must also take care that the body is cleaned and that the clothing is as intact and decent as possible [19,23,46]. Good practices for death notification in the emergency department should also ensure that survivors spend as little time as possible in the hospital [17]. 

The emergency room is generally a frenetic environment, full of emotions, where roles can often be confused and overlap, the staff may appear more “insensitive”, and the waits are exhausting [23]. Furthermore, doctors working in these departments have limited time availability; they also must devote themselves to other patients, and it is possible that they are not always present due to work shifts [18]. For this reason, it is convenient that the doctor is supported by an expert staff member (e.g., a nurse), who has the task of accompanying the relatives through the notification process, from the reception to the viewing of the body, up to the practical procedures and post-notification follow-up [1,14,19,37,44].

The presence of at least two staff members is essential when death notification is carried out by members of law enforcement, to the point that a team approach is widely encouraged [31,50,59,64]. The presence of two or more officers ensures constant intervention for the relatives who receive the notification and, at the same time, for those who are preparing to call the other family members and organize subsequent actions (e.g., transport to the hospital).

One study analyzed the role of forensic nurses as special reference figures to assess the needs of survivors and offer an effective support plan even post-notification [50]. Cooperation between the different roles assumed by police personnel is a fundamental element for the success of the notification process [31]. Survivors need to be able to have the name and telephone number of the investigating officer conducting the investigation, receive the right information, and be informed of the actions that any ongoing investigation will involve, especially in the case of homicide [59].

In summary, it is necessary to draw up best practices for the notification of death in relation to the professional figures in charge of the act. In the hospital setting, the literature highlights the need for an adaptation of the role of the professional, from the treatment of the medical condition to the treatment of the family trauma, from protector to listener, with the possibility of being assisted by a competent staff member throughout the notification process.

With respect to training, this review has highlighted that, in most of the contexts, the staff had never been trained to deliver bad news or had received only partial, inconsistent, and/or inadequate training to meet the needs of the context. The need for training emerges as a relevant issue in relation to various professional figures, both with respect to the deed and the entire notification process, especially for law enforcement agencies, of which very few studies are available in the literature.

To date, existing death notification training is being formalized through different forms of psychoeducation coupled with exercises, simulations, role-plays, and small group discussions. The vast heterogeneity of approaches has not made it possible to identify a common training model for teaching activities in the field, not even by focusing the intervention on specific professional categories. This limitation also hinders the possibility of carrying out an evaluation of effectiveness by comparing the different interventions.

Following the COVID-19 pandemic, several studies have adapted the training modules to e-learning methods, expanding the usability of the training offered and obtaining consistent and reliable internal effectiveness assessments. Using remote modalities is also an integral part of death notification best practices. Although initial contact by telephone is common to all notification protocols, this phase of the notification process is understudied. It is necessary that, from the first contact, especially since it takes place remotely, the notifier assume an empathic, calm, and controlled attitude. The caller should make sure he understands where the person is, his emotional state, and whether he has someone nearby who can support him and accompany him to the predefined location. In these terms, the recent CONNECT protocol [68], formulated following the COVID-19 pandemic, offers important logistical and organizational cues for delivering bad news remotely.

The last area examined concerns the best practices for notification of death to children, an area underexplored but of vital importance. In fact, it can often happen that the notifiers must interface with families in which there are minors who, by virtue of their fragility, need a careful and diversified approach. Specifically, it is essential to train professionals in direct communication styles with children, in assessing the suitability of the context and available support resources, and in providing adults with advice and practical actions to take if they are responsible for making the notification.

The analysis of the literature suggests that death notification should be considered both an act and a process, which includes different phases. The construction of new specific guidelines for the communication of traumatic death, as well as the adoption of existing protocols, should consider the needs of the actors involved and the context in which the notification occurs. In particular, the notification of traumatic deaths to children requires a specific and sensitive approach. The professionals should assist and support the parents in this difficult task, providing practical skills and strategies. The communication of traumatic death by telephone is a new and little-explored topic. On a practical level, studies suggest that the focus of attention should be on the safety of the recipient; in addition, they highlight the importance of providing follow-up contact. Appropriate training on the task of death notification, providing both theoretical knowledge and practical abilities, should be included in the educational path of professionals in order to reduce the negative impact of the notification act.

## 5. Limitations

This review has several limitations. Firstly, there was no review protocol. Secondly, only studies written in English were considered. Thirdly, between-study heterogeneity was high with respect to study design, quality, and study objectives, making it quite difficult to create standardized categorizations for all articles. Furthermore, thematic overlaps were frequent, with most studies reporting general guidance on all important areas of the topic, sometimes succinctly, sometimes quite extensively. These limitations, combined with the difficulty—on some occasions—in distinguishing between guidelines and training programs, created difficulties in interpreting the results; for example, it was not easy to obtain clear-cut indications for specific circumstances. Furthermore, the scarcity of quantitative studies does not allow for the identification of the advantages of some types of intervention over others.

## 6. Conclusions

The analysis of the existing literature on the notification of sudden, violent, and traumatic deaths has highlighted the importance of promoting guidelines that accompany the professional throughout the notification process, from the first contact to the follow-up phase. An examination of the material produced on death communication reveals the need to implement research in this area and to create specific protocols for the characteristics and needs associated with the various professional contexts. In particular, the indications addressed to law enforcement officers are still limited, and few studies have paid attention to this category of professionals [3,31,50,59,64].

A further area of interest for future studies concerns the notification of death to children, which must be carried out in a protected context, taking into consideration their vulnerabilities, and assisting adults in this difficult task [14,39].

Finally, the literature has highlighted the need to encourage training, which must take place in standardized contexts with customized programs for the different contexts and professional categories and must provide for the combination of theory through the teaching of psychoeducational aspects and practice through the introduction of role-playing, exercises, and simulations. Indeed, future research needs a narrower focus in order to ensure that the evidence base takes into account the lived experience of the bereaved and organizational-specific settings.

## Figures and Tables

**Figure 1 ijerph-20-06222-f001:**
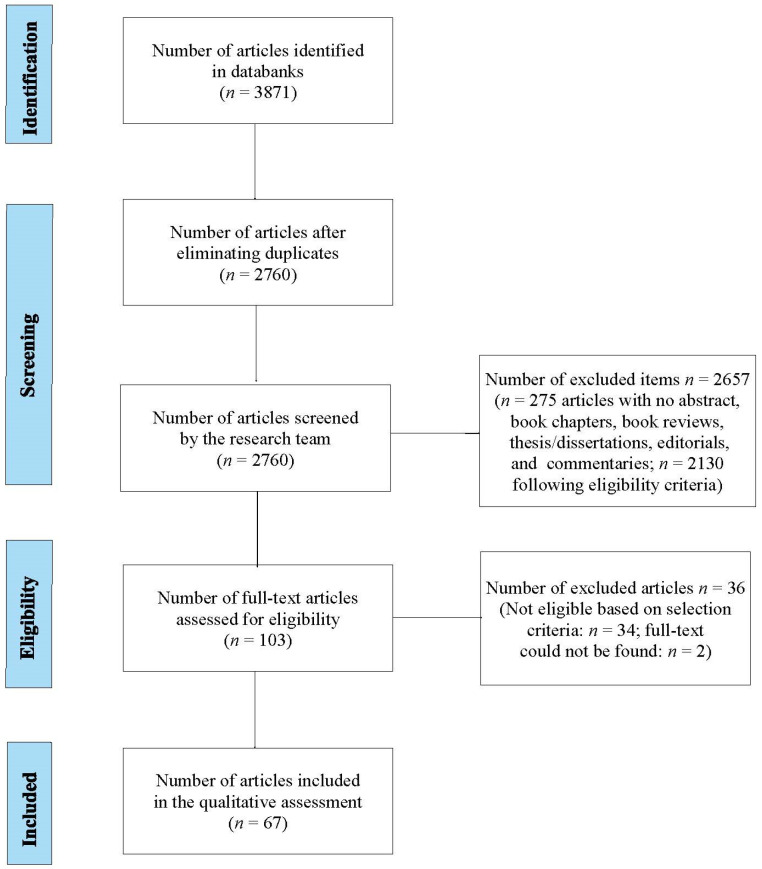
Decision tree for study selection based on PRISMA criteria.

**Figure 2 ijerph-20-06222-f002:**
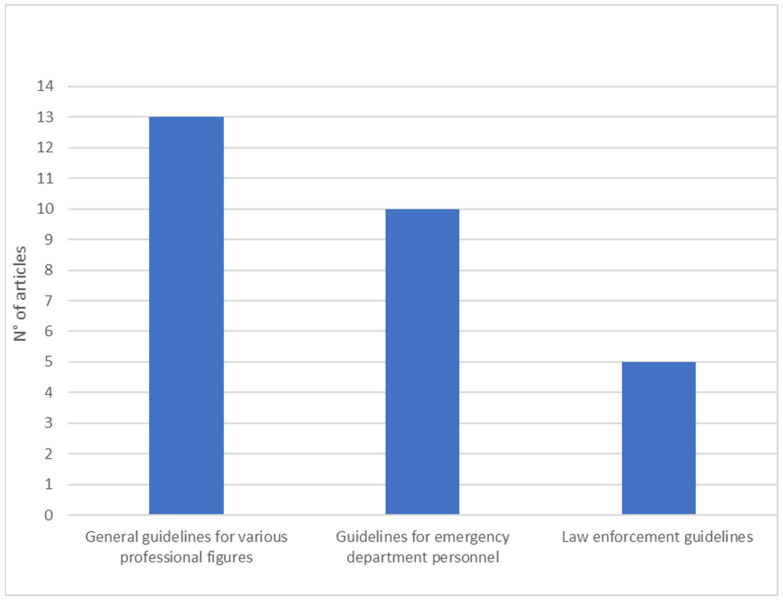
General guidelines and guidelines in relation to various professional figures.

**Figure 3 ijerph-20-06222-f003:**
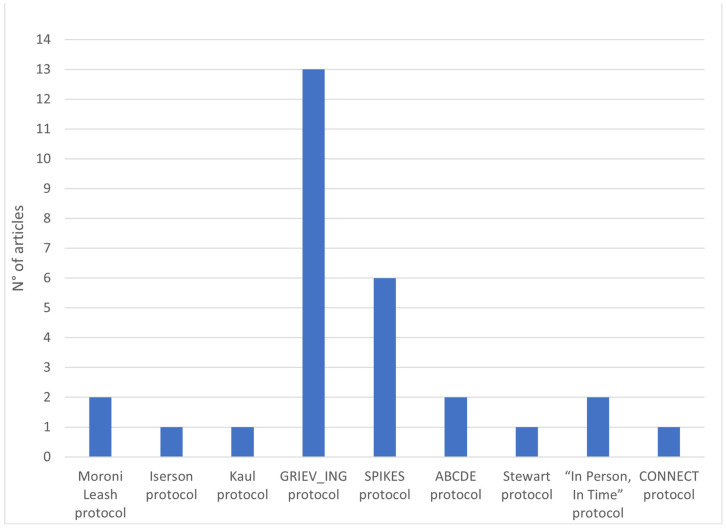
Articles reporting on specific protocols.

**Figure 4 ijerph-20-06222-f004:**
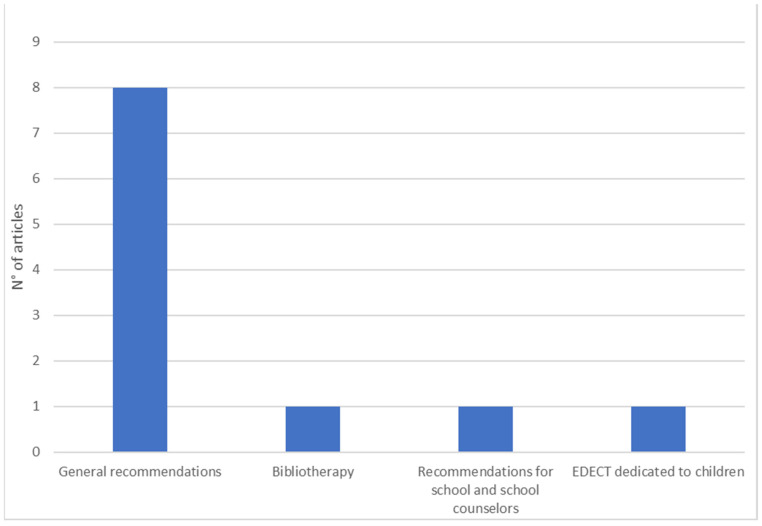
Guidelines for death notification to children.

**Figure 5 ijerph-20-06222-f005:**
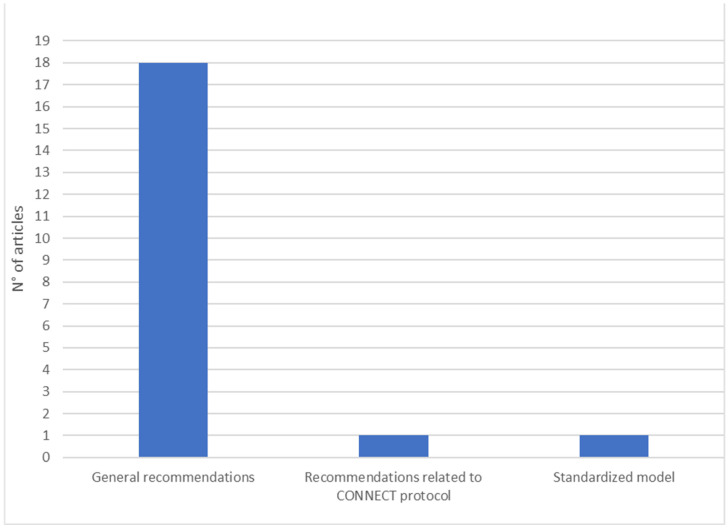
Guidelines for death notification by phone.

**Figure 6 ijerph-20-06222-f006:**
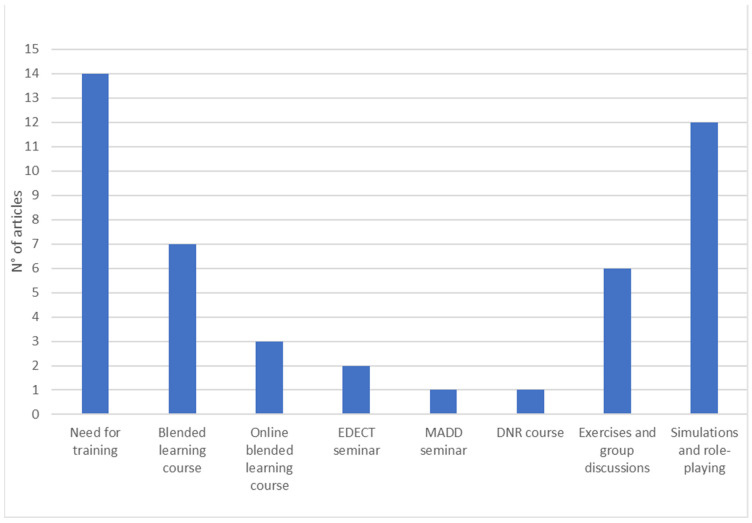
Good practice for death notification training programs.

**Table 1 ijerph-20-06222-t001:** Characteristics of studies included in the systematic review.

Author (Year)	Country	Target Population	Sample (*n*)	Study Design	Dimensions				
					General Guidelines, and in Relation to Various Professional Figures	Specific Protocols	Guidelines for Death Notification to Children	Guidelines for Death Notification by Phone	Good Practice for Death Notification Training Programs
Dubin and Sarnoff (1986) [14]	Pennsylvania (USA)	Doctors and health personnel of emergency departments.	NA	Descriptive/ narrative	Recommendations for the emergency department.	NA	Recommendations	Recommendations	NA
Spencer et al. (1987) [15]	California (USA)	Homicide detectives of the Los Angeles Police Department.	Fifty subjects filled out a questionnaire. Twenty-one of them also made a telephone interview.	Cross-sectional survey.	NA	NA	NA	NA	Need for training.
Schmidt et al. (1992) [16]	Oregon (USA)	Residents of Oregon Health Sciences University.	NA	Descriptive/narrative.	NA	NA	NA	Recommendations	Blended learning course for residents in the emergency department.
Adamowski et al. (1993) [17]	Canada	Two groups of survivors in Ottawa General Hospital.	Sixty-six subjects.	Survey.	Recommendations for the emergency department.	NA	NA	Recommendations	NA
Swisher et al. (1993) [18]	Pennsylvania (USA)	Forty-five residents and twenty physicians in emergency departments at the Medical College of Pennsylvania.	Sixty-five Subjects.	Survey.	Recommendations for the emergency department.	NA	NA	NA	Need for training.
Marrow (1996) [19]	UK	Emergency department personnel.	NA	Descriptive/ narrative.	Recommendations for the emergency department.	NA	Recommendations.	NA	Need for training.
Moroni Leash (1996) [20]	California (USA)	Medical professionals, university students in death and dying classes, and family members of newly admitted intensive care unit patients.	Two hundred medical professionals, 100 university students, and 100 family members.	Descriptive/ narrative.	NA	Moroni Leash protocol	Recommendations	Recommendations	NA
Von Bloch (1996) [21]	Texas (USA)	Health care professionals.	NA	Descriptive/narrative.	Recommendations for health professionals.	NA	NA	NA	NA
Ahrens and Hart (1997) [22]	Illinois (USA)	General emergency physicians.	One hundred and twenty-two subjects.	Survey	NA	NA	NA	NA	Need for training.
Olsen et al. (1998) [23]	Chicago (USA)	Emergency department personnel.	NA	Descriptive/narrative.	Recommendations for physicians in the emergency department.	Recommendations.	NA	NA	NA
Smith et al. (1999) [24]	Maryland (USA)	Emergency physicians, paramedics, and other emergency personnel.	NA	Descriptive/narrative.	NA	NA	NA	NA	Seminar: EDECT.
Stewart (1999) [25]	Florida (USA)	Those involved in notifying a road accident-related death.	NA	Descriptive/narrative.	Recommendations in case of road accidents.	NA	NA	Recommendations.	NA
Kaul (2001) [26]	Michigan (USA)	Emergency physicians, paramedics, and other emergency personnel.	NA	Descriptive/ narrative.	NA	Iserson protocol; Moroni Leash protocol; and Kaul protocol.	NA	Recommendations.	NA
Stewart et al. (2001) [27]	Florida (USA)	Death notifiers (law enforcement officers, emergency medical technicians, victim advocates, coroners, etc.) in seven cities of the USA.	Two hundred and forty-five subjects.	Survey.	NA	NA	NA	Recommendations.	Need for training; MADD seminar.
Benenson and Pollack (2003) [28]	England	Emergency medicine residents.	Seventy subjects.	Prospective observational.	NA	NA	NA	NA	Blended learning course.
Janzen et al. (2003–2004) [29]	Canada	Parents who had experienced the sudden death of a child in Ontario.	Twenty subjects.	Qualitative study.	Recommendations for various professionals in cases of pediatric death.	NA	NA	NA	NA
Servaty-Seib et al. (2003) [30]	Indiana (USA)	School communities.	NA	Descriptive/ narrative.	NA	NA	Recommendations for school and school counselors.	NA	NA
Hart and DeBernardo (2004) [31]	Baltimore (USA)	Law enforcement personnel.	NA	Descriptive/ narrative.	Recommendations for police officers.	NA	NA	Recommendations.	NA
Levetown (2004) [32]	Texas (USA)	Emergency care personnel.	NA	Descriptive/ narrative.	Recommendations for staff of the emergency department in the event of the death of a child.	NA	NA	Recommendations.	NA
Hobgood et al. (2005) [33]	North Carolina (USA)	Residents in emergency medicine.	Twenty subjects.	Pre-post study.	NA	GRIEV_ING protocol.	NA	NA	NA
Smith-Cumberland (2005) [34]	Maryland, Pennsylvania, and Utah (USA)	Emergency medical technicians from 14 states.	One hundred and thirty-six subjects.	Survey.	NA	NA	NA	NA	Need for training.
Eberwein (2006) [35]	Maryland (USA)	Mental health professionals.	NA	Descriptive/ narrative.	Recommendations for various professionals.	NA	NA	Recommendations.	NA
Smith-Cumberland and Feldman (2006) [36]	Wisconsin (USA)	Emergency medical professionals.	Eighty-three subjects.	Pre-post study.	NA	NA	Part of EDECT dedicated to children.	NA	Blended learning course for emergency department: EDECT, with a focus on CME units.
Scott (2007) [37]	England	Emergency medical professionals and police officers.	NA	Descriptive/narrative.	Recommendations for the staff of the emergency department.	NA	NA	NA	NA
Miller (2008) [38]	Florida (USA)	Professionals potentially involved in notifying death.	NA	Descriptive/narrative.	Recommendations for various professionals in the case of murder.	NA	Recommendations.	Recommendations.	NA
Mitchell (2008) [39]	Maryland (USA)	Professionals potentially involved in notifying death.	NA	Descriptive/ narrative.	Recommendations for various professionals.	NA	Recommendations.	Recommendations.	NA
Hobgood et al. (2009) [40]	North Carolina (USA)	Fourth year medical students at the University of North Carolina.	One hundred and forty-eight subjects.	Pre-post study.	NA	GRIEV_ING protocol.	NA	NA	NA
Park et al. (2010) [41]	Florida (USA)	Emergency medicine residents.	Fourteen subjects.	Descriptive/narrative.	NA	SPIKE protocol; GRIEV_ING protocol.	NA	NA	Blended learning course for the emergency department.
Parris (2011) [1]	UK	Emergency medical professionals.	NA	Descriptive/ narrative.	Recommendations for the emergency department.	NA	NA	Recommendations.	NA
Douglas et al. (2012) [42]	Canada	Paramedics in urban and rural areas of Ontario.	Twenty-eight subjects.	Qualitative study.	NA	NA	NA	NA	Need for training.
Marco and Wetzel (2012) [43]	Ohio (USA)	Patients who were involved in a fatal motor vehicle crash between 2005 and 2009.	Twenty-one subjects.	Cross-sectional survey study.	Recommendations in the case of a motor vehicle crash.	NA	NA	NA	NA
Roe (2012) [44]	Michigan (USA)	Emergency medical professionals.	NA	Descriptive/ narrative.	Recommendations for the staff of the emergency department.	NA	NA	NA	NA
Shaw et al. (2012) [45]	Australia	Doctors employed in Sydney metropolitan hospitals.	Thirty-one subjects.	Mixed-method design (quantitative/ qualitative).	NA	NA	NA	NA	Need for training.
Hobgood et al. (2013) [6]	North Carolina (USA)	Emergency medical service personnel.	Thirty subjects.	Pre-post design.	NA	GRIEV_ING protocol.	NA	NA	NA
Scott (2013) [46]	England	Emergency medical professionals.	NA	Descriptive/ narrative.	Recommendations for the staff of the emergency department.	NA	NA	NA	Exercises for residents.
Shoenberger et al. (2013) [47]	South California (USA)	Physicians of emergency departments.	NA	Review.	NA	GRIEV_ING protocol.	NA	NA	Need for training.
Shomoossi et al. (2013) [7]	Iran	Nurses working in hospitals in Sabzevar, in Iran.	Ninety-seven subjects.	Development and validation of a scale.	NA	SPIKES protocol; ABCDE protocol; and Stewart protocol.	NA	NA	NA
Sobczak (2013) [8]	Poland	Doctors involved in death notification.	NA	Descriptive/ narrative.	NA	SPIKES protocol; “In person, in time” protocol.	.	Recommendations.	NA
Galbraith et al. (2014) [48]	Midwestern (USA)	Senior-level nursing and social work students.	Thirty-two subjects.	Development of a valid simulation model.	NA	NA	NA	NA	Simulation for nursing and social work students.
Rivolta et al. (2014) [49]	Torino (Italy)	Health care nurses from two nursing homes and two hospices.	Fifty-five subjects.	Qualitative study.	Recommendations for staff of hospices and nursing homes.	NA	NA	NA	NA
Baumann and Stark (2015) [50]	New Jersey (USA)	Forensic death investigators and other death notifiers.	NA	Descriptive/narrative.	Recommendations for forensic nurses (FNDIs).	NA	NA	NA	NA
Reed et al. (2015) [51]	Ohio (USA)	First year pediatric and internal medicine residents.	Forty-four subjects.	Pre-post design.	NA	GRIEV_ING protocol.	NA	NA	NA
Basinger et al. (2016) [52]	Midwestern (USA)	College students who had lost a parent or a sibling.	Twenty-one subjects.	Qualitative study.	Privacy management process.	NA	Recommendations.	NA	NA
Carmack and DeGroot (2016) [53]	Virginia (USA)	Lay people recruited via social media and other means.	Three hundred and two subjects in study 1; three hundred and nineteen in study 2.	Development and validation of a new scale.	NA	NA	NA	NA	Need for training.
Veilleux and Bilsky (2016) [54]	Arkansas (USA)	Therapists and residents after a patient’s death (e.g., suicide).	NA	Descriptive/narrative.	Recommendations in the case of the suicide of a patient.	NA	NA	Recommendations.	NA
Arruda-Colli et al. (2017) [55]	Maryland (USA)	Storybooks about dying, death, and bereavement in English, French, or Spanish, published 1995–2015.	Two hundred and ten subjects.	Review article.	NA	NA	Bibliotherapy.	NA	NA
Brand and Mahlke (2017) [11]	Germany	German police officers.	NA	Descriptive/ narrative.	NA	NA	NA	NA	DNR—blended learning course.
Karam et al. (2017) [12]	Lebanon	Residents of PGY3 and PGY4 Lebanese anesthesiology.	Sixteen subjects.	Pre-post training.	NA	GRIEV_ING protocol.	NA	NA	NA
Ombres et al. (2017) [56]	Maryland (USA)	Internal medicine residents.	Sixty-seven subjects.	Review article.	NA	NA	NA	Standardized model.	NA
Shakeri et al. (2017) [57]	Chicago (USA)	American university—4 year emergency medicine training program.	Forty subjects.	Validation study.	NA	GRIEV_ING protocol.	NA	NA	NA
Williams-Reade et al. (2018) [58]	California (USA)	Pediatric surgery residents.	15	Pre-post study.	NA	SPIKES protocol.	NA	NA	Seminars and simulation.
Reed et al. (2019) [59]	Georgia (USA)	Homicide unit members/homicide co-victims.	52 (26 + 26)	Qualitative study.	Recommendations for police officers.	NA	NA	Recommendations.	Need for training.
Servotte et al. (2019) [60]	Belgium	Medical students and residents.	68	Pre-post study.	NA	SPIKES protocol.	NA	NA	Blended learning course.
Ahmed et al. (2020) [10]	Indiana (USA)	Emergency medicine residents.	22	Pre-post study.	NA	GRIEV_ING protocol.	NA	NA	Simulation and Rapid Cycle Deliberate Practice (RCDP).
Campos et al. (2020) [13]	Nebraska/Massachusetts/Ohio/New York/Michigan/Texas (USA)	Emergency medical services professionals.	1514	Survey	NA	NA	NA	NA	Need for training.
De Leo et al. (2020) [5]	Italy	NA	Sixty articles.	Systematic review.	Recommendations for various professionals.	GRIEV_ING protocol; SPIKES protocol; ABCDE protocol; and “In Perso, In Time” protocol.	Recommendations.	Recommendations.	Blended learning course.
Hughes et al. (2020a) [61]	Michigan (USA)	Medical school students.	12	Validation of a digital communication platform model.	NA	GRIEV_ING protocol.	NA	NA	Blended learning course (digital platform).
Hughes et al. (2020b) [62]	Florida/Indiana/Arizona/Michigan (USA)	Emergency medicine physicians.	NA	Descriptive/narrative.	NA	GRIEV_ING protocol	NA	NA	Simulation and Rapid Cycle Deliberate Practice (RCDP).
Landa-Ramìrez et al. (2020) [63]	Mexico	Health care professionals.	NA	Descriptive/narrative.	NA	NA	NA	Recommendations.	Online blended learning course.
Fiorentino et al. (2021) [9]	New Jersey (USA)	Trauma (general surgery and EM) residents.	39	Pre-post study.	NA	NA	NA	NA	Blended learning course.
Hofmann et al. (2021a) [64]	Germany	Police officers.	142	Pre-post study.	NA	NA	NA	NA	Blended learning course (e-learning program) for police officers.
Hofmann et al. (2021b) [2]	Germany	Police officers.	86	Qualitative study.	Recommendations for police officers.	NA	NA	NA	Need for training for police officers.
Hughes et al. (2021) [65]	Michigan/Indiana (USA)	Emergency medicine residents/ medical school students.	Thirty-four (22 residents + 12 students).	Pre-post study.	NA	GRIEV_ING protocol.	NA	NA	Simulation and Rapid Cycle Deliberate Practice (RCDP).
De Leo et al. (2022a) [66]	Italy	Police officers/ health professionals (doctors and nurses).	155	Qualitative study.	Recommendations for various professionals.	NA	NA	NA	Need for training.
De Leo et al. (2022b) [67]	Italy	Survivors of traumatic death	52	Qualitative study.	Recommendations for various professionals.	NA	NA	NA	NA
McGill et al. (2022) [3]	United Kingdom	Survivors (spouses, parents, and children) of the deaths of UK Armed Forces members.	15	Qualitative study.	Recommendations for veterans.	NA	Recommendations.	NA	NA
Sobczak (2022) [68]	Poland	Health care professionals.	NA	Development of a protocol for remote communication of bad news.	NA	CONNECT protocol.	NA	Recommendations related to the CONNECT protocol.	NA

## Data Availability

No new data were created or analyzed in this study. Data sharing is not applicable to this article. Publicly available datasets were analyzed in this study. This data can be found here: EBSCO PsycInfo https://web.s.ebscohost.com/ehost/search/advanced?vid=0&sid=3e466d57-bf54-4246-8f9c-3ea90435109c%40redis; EBSCO CINAHL https://web.s.ebscohost.com/ehost/search/advanced?vid=5&sid=8eecb5ea-31dc-4b40-b315-d4c7ecd3a798%40redis; Scopus https://www.scopus.com/search/form.uri?display=basic#basic; MEDLINE PubMed https://pubmed.ncbi.nlm.nih.gov/; Web of Science https://www.webofknowledge.com; accessed on 5 November 2022.

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
