# Peer review of "Best Practices for Notification of Unexpected, Violent, and Traumatic Death: A Scoping Review"

_ijerph, 2023, doi:10.3390/ijerph20136222_

Round 1
Reviewer 1 Report
This is a very interesting concept and valuable as an exploratory study. What is missing is the definition of traumatic death/sudden death, both terms you use in conjunction with each other. This is a very broad subject area and it is not clear if you are relating this to clinical settings only, emergency services, Armed Forces or every situation where traumatic death might occur. You also mention children, which is, or should be, considered separately to informing and support adults who are bereaved. The methods section, your literature search and the rigour, is good. Not sure this is the best methodology for transactional research that seeks to support shaping how news of a traumatic death is delivered in the future.
Reviewer 2 Report
Thank you for this interesting scoping review. It provides ideas for improvement of practice. The only think i would suggest to change is to provide a table with a summary of the most important findings and suggestions to improve practice and research.
Reviewer 3 Report
Dear Editor,
I appreciate the opportunity to review the article “Best Practices for Notification of Unexpected, Violent and Traumatic Death: A Scoping Review”. An interesting work whose objective is referred:To provide an overview of the literature on best practices in the task of notification of 15 sudden, violent and traumatic death; to provide guidance for the formulation of appropriate best 16 practices and the development of effective educational programs.
In this topic, I would suggest changing the objective to Map what is produced on Best Practices for Notification of Unexpected, Violent and Traumatic Death.
The subject of the study is very relevant , I just suggest some revisions:
- 1966 to 2018, Why? These were the results he obtained. The correct is to say that the research was conducted without a time limit.
- Point out in the introduction, that no other study, review or protocol on this topic was identified.
- Refer in the method that no register was made,
- The dimensions identified are: (1) general guidelines, in relation to various profes-133 sional figures (n = 28); (2) specific protocols (n = 21); (3) guidelines for notifying children 134 of death (n = 11); (4) guidelines for telephone death notification (n = 21); (5) recommenda-135 tions and suggestions for death notification training programs (n = 32) (see Table 1) (table 1 is missing) . I suggest presenting a results diagram
- This part put in the results is method:
In particular, the identification of the dimensions followed a phase of reflective and 124 deductive thematic analysis. Once data collection was complete, four researchers on the 125 team (JZ, AVG, MC, GM) individually performed the analysis steps, during which they 126 took notes on their initial impressions of each article. In a second step, the contents of 127 interest (i.e., those in line with the research question) were assigned labels (a few words 128 or a short sentence), which had the purpose of clearly evoking the relevant characteristics 129 of the papers, in order to be able to encode them. Then, the researchers - in full agreement 130 with each other - defined a list of themes, which concluded in five dimensions that guided 131 the subsequent research phases.
-I suggest reducing the data presented in the results.
Round 2
Reviewer 1 Report
Thank you for revising the paper and addressing the comments. This is an important subject area. It remains a broad scope and, therefore, adopting good practice does need to have a caveat that reflects this. You have strengthened this important point that practice will depend on the setting and organisation type. And, the methodological approach is better explained. To strengthen this further, it is worth adding that future research needs a narrower focus in order to ensure that the evidence-base takes into account lived experience of the bereaved and organisational specific settings.
